# Bacterial Toxins Active against Mosquitoes: Mode of Action and Resistance

**DOI:** 10.3390/toxins13080523

**Published:** 2021-07-27

**Authors:** Maria Helena Neves Lobo Silva-Filha, Tatiany Patricia Romão, Tatiana Maria Teodoro Rezende, Karine da Silva Carvalho, Heverly Suzany Gouveia de Menezes, Nathaly Alexandre do Nascimento, Mario Soberón, Alejandra Bravo

**Affiliations:** 1Department of Entomology, Instituto Aggeu Magalhães-FIOCRUZ, Recife 50740-465, PE, Brazil; tatiany.melo@fiocruz.br (T.P.R.); tatiana.rezende@fiocruz.br (T.M.T.R.); karine.carvalho@fiocruz.br (K.d.S.C.); suzany.menezes@cpqam.fiocruz.br (H.S.G.d.M.); 2Departament of Molecular Microbiology, Instituto de Biotecnología, Universidad Nacional Autónoma de México (UNAM), Cuernavaca 62250, MR, Mexico; nathaly.nascimento10@gmail.com (N.A.d.N.); mario.soberon@ibt.unam.mx (M.S.); alejandra.bravo@ibt.unam.mx (A.B.)

**Keywords:** Bti, *Lysinibacillus sphaericus*, cry toxin, Cyt toxin, binary toxin, receptors

## Abstract

Larvicides based on the bacteria *Bacillus thuringiensis* svar. *israelensis* (Bti) and *Lysinibacillus sphaericus* are effective and environmentally safe compounds for the control of dipteran insects of medical importance. They produce crystals that display specific and potent insecticidal activity against larvae. Bti crystals are composed of multiple protoxins: three from the three-domain Cry type family, which bind to different cell receptors in the midgut, and one cytolytic (Cyt1Aa) protoxin that can insert itself into the cell membrane and act as surrogate receptor of the Cry toxins. Together, those toxins display a complex mode of action that shows a low risk of resistance selection. *L. sphaericus* crystals contain one major binary toxin that display an outstanding persistence in field conditions, which is superior to Bti. However, the action of the Bin toxin based on its interaction with a single receptor is vulnerable for resistance selection in insects. In this review we present the most recent data on the mode of action and synergism of these toxins, resistance issues, and examples of their use worldwide. Data reported in recent years improved our understanding of the mechanism of action of these toxins, showed that their combined use can enhance their activity and counteract resistance, and reinforced their relevance for mosquito control programs in the future years.

## 1. Entomopathogenic Bacteria Active against Mosquito Larvae

Insects can act as vectors of etiological agents of different diseases and can be a nuisance to humans, being responsible for health burdens worldwide [1]. Re-emergent and emergent diseases, in particular arboviruses, remain a global challenge as recently shown for the epidemic problems caused by the Zika virus [2]. Microbial larvicides based on entomopathogen bacteria have been successfully used for controlling mosquito and black-fly populations, as an alternative to the conventional classes of chemical insecticides, due to their high effectiveness and environmental safety [3,4,5]. *Bacillus thuringiensis* serovariety (svar.) *israelensis* (Bti) de Barjac was the first *B. thuringiensis* (Bt) bacterial serotype identified as active against some Diptera larvae [6]. Subsequently, *Lysinibacillus sphaericus* (*L. sphaericus*) Neide strains, with activity against Culicidae larvae were discovered [7]. Since the 1980s, products based on these two bacteria have been considered the most successful biological agents for controlling the larvae of mosquitoes and black-flies [5,8,9,10]. Bti and *L. sphaericus* are Gram-positive, aerobic, sporulating, and cosmopolitan bacteria that display high and selective larvicidal activity against Diptera including genera of public health importance such as *Aedes*, *Anopheles, Culex*, and *Simulium* [5]. In this review we will focus on describing the recent knowledge regarding to the mechanism of action of the insecticidal proteins produced by these bacteria and their synergism, and examples of utilization for mosquito control, since another review in this Special Issue will cover bacterial toxins in the control of dipteran insect pests of medical and agronomic importance [11].

The larvicidal activity of Bti and *L. sphaericus* is due to the production of crystalline inclusions during their sporulation phase of growth (Figure 1). These crystals are composed of protoxins that act on the midgut epithelial cells of the larvae after ingestion, targeting specific membrane-bound receptors [12,13]. Notably, the selective mode of action of these toxins is the major feature considered in the actual safety requirements for larvicides employed for mosquito control. According to the classification of the Insecticide Resistance Action Committee (https://irac-online.org/, accessed on 20 June 2021), those toxins belong to the mode of action Group 11 designed as “Microbial Disruptors of Insect Midgut Membranes”. In order to be active, the insecticidal crystals produced by Bti and *L. sphaericus* are required to be ingested by the larvae. Inside the gut, the protoxins are processed into active toxins that finally bind to midgut receptors, leading to pore formation in the midgut cell membranes [9,14].

Bti was discovered by Margalit and Goldberg in 1977 [15] and classified by its flagellar serotyping as H-14 [6], remaining as a reference strain. The full classification of *Bacillus thuringiensis* (Bt) serovars and their toxins [16] can be found in http://www.lifesci.sussex.ac.uk/home/Neil_Crickmore/Bt/ accessed on 18 June 2021. The insecticidal parasporal crystal produced by Bti is commonly composed of four major protoxins [17] with a selective spectrum that includes the larvae of Culicidae (mosquitoes), Simuliidae (black-flies), and Chironomidae (midges) species. The major advantage of Bti is its proven field effectiveness after more than four decades of use, without records of larvae resistance to the insecticidal crystal [4,5,18].

The first mosquitocidal strain of *L. sphaericus,* formerly classified as *Bacillus sphaericus* [19], was isolated by Kellen et al. [7], but the first strains studied, such as the Kellen strain, showed low toxicity. Later, the discovery of additional strains displaying high larvicidal activity—such as 1593 [20,21], 2362 [22], and C3-41 [21]—led to the development of commercial larvicides [13]. *L. sphaericus* isolates were grouped into flagellar serotypes by DNA homology analysis [13,23], and recent genomic sequencing has also contributed to their classification [24]. Among the insecticidal factors that were characterized in *L. sphaericus* strains, the crystal containing the binary (Bin) protoxin is by far the most important. Recently, a novel nomenclature of pesticidal proteins based on their protein structure named the Bin toxin as “Tpp” toxin pesticidal protein [16], while the Cry mnemonic was retained for the three domain proteins, and the Cyt mnemonic was retained for the Cyt-related proteins [16]. The Bin crystal of *L. sphaericus* has a narrow spectrum of action compared to the Bti’s crystal, as it only targets Culicidae larvae [5]. Field performance in breeding sites with organically polluted water is an outstanding feature of Bin crystals, but insects resistant to the Bin crystal have evolved causing a problem that requires additional management practices [25,26,27]. This review aims to present the major features and recent knowledge of Bti and *L. sphaericus* mosquitocidal toxins and the opportunities to exploit them based on novel advances regarding their specific mode of action in the midgut cells and on the plethora of experiences derived from their field utilization.

## 2. Toxins and Mode of Action

### 2.1. Bacillus thuringiensis *svar.* israelensis (Bti)

A few years after its discovery, Bti-based larvicides were introduced for vector control, and to date, this biological control strategy remains effective and safe. The multi-toxin composition of Bti crystal and its complex mode of action play an important role to provide their selective action associated with the lack of insect resistance to the crystal [3,12]. During the sporulation phase, this bacterium produces Cry and Cyt insecticidal protoxins that accumulate in parasporal crystals. The genes that code for those toxins are located in a 128 kb pBtoxis megaplasmid where the main protoxins are Cry4Aa (125 kDa), Cry4Bb (135 kDa), Cry11Aa (68 kDa), and Cyt1Aa (28 kDa) [17]. Some strains can also produce lower quantities of additional protoxins such as Cry10Aa (58 kDa) and Cyt2Ba (30 kDa), which also display toxicity against mosquito larvae [28,29,30]. The individual Cry and Cyt proteins from Bti show low toxicity to mosquito larvae, compared to the high toxic effect displayed by the whole Bti crystal, which results from the synergism among these proteins [31]. The first major steps described in the Bti’s crystal mode of action are their ingestion by mosquito larvae, the crystals’ solubilization in the alkaline pH of the midgut lumen, and the protoxin activation by midgut proteases ending in pore formation into the midgut cells [32,33,34]. The ingestion of the crystals is important for the mode of action since it was observed that larvae treated with soluble toxins did not display mortality [35]. Both Cry and Cyt are pore-forming toxins that destroy the epithelium midgut cells causing larval death. The production of different toxins with distinct modes of action is a key feature since Cry toxins rely on a variety of protein receptors, while Cyt toxins bind directly to the membrane lipids. Regarding the interaction with receptors, it is worth noting that the Cyt1Aa toxin acts as a surrogate receptor for the Cry toxins as described below [12,33,36].

#### 2.1.1. Cry Toxins

The Cry4Aa, Cry4Ba, and Cry11Aa toxins are composed of three domains: domain I is involved in toxin oligomerization and in the pore-formation activity, while domains II and III are involved in receptor binding [9]. The available crystallographic structures of Cry4Ba and Cry4Aa showed their three-domain structure [34,37,38,39,40,41,42]. The proposed model of their mode of action is the pore-formation model that was first established for Cry1A toxins in the midgut of the lepidopteran *Manduca sexta* and involves the sequential binding of the toxin to different receptors. First, the monomeric Cry toxin binds with low affinity to the highly abundant glycosylphosphatidylinositol (GPI)-anchored receptors such as aminopeptidases (APN) and alkaline phosphatases (ALP); then, the toxin binds to cadherin (CAD), which is a transmembrane protein, with higher affinity. This interaction induces the cleavage of the helix α-1 promoting oligomer formation. The Cry oligomers bind with higher affinity to APN or ALP, and it is proposed that this interaction is needed to insert the oligomer into the cell membrane, forming pores that cause osmotic shock and kill the larvae [43,44,45,46]. In the case of mosquitoes, the Cyt1Aa toxin may act as an additional receptor for Cry toxins, promoting their oligomerization and membrane insertion [47], as summarized in Figure 2. The oligomerization of Cry11Aa and Cry4Ba is an essential step for their toxicity, and it was shown that helix α-3 of domain I is involved in this step [48,49]. It was also demonstrated that the binding of Cry11Aa to CAD is required for its toxic action. However, Cry4Ba after proteolytical activation can oligomerize in the absence of this receptor [50].

The binding interactions of the Cry toxins from Bti with GPI-anchored receptors from the midgut epithelium are important for their mode of action [51]. For the Cry11Aa toxin, the receptors described in *Ae. aegypti* are ALP1, AaeAPN1, AaeAPN2, and AaeCAD [52,53,54,55,56,57,58,59]. A GPI-anchored α-amylase (Aamy1) was also identified as a receptor for the Cry11Aa toxin in *Anopheles albimanus* [60]. Another study suggested that Cry11Aa toxicity in *Ae. aegypti* also depends on an ATP-binding cassette protein [61], although more studies are necessary to determine the role of this molecule as a Cry toxin receptor in mosquitoes. The receptors characterized for the Cry4Ba toxin in *Ae. aegypti* are the proteins APN (2778, 2783, and 5808) and ALP (ALP1 and *Aa*-mALP) [54,62,63,64]. CAD proteins (AgCAD1 and BT-R3) were also identified as receptors of the Cry4Ba toxin in *Anopheles gambiae* [65,66]. The involvement of ALP in Cry4Aa toxicity was demonstrated when *Ae. aegypti* larvae with reduced ALP expression showed increased survival after being exposed to this toxin [67]. Some important regions involved on the binding between Cry toxins and their midgut receptors were identified [68,69,70,71]. It was previously reported that the APN, ALP, and CAD receptors are located on the epithelial cells from the caeca and posterior midgut, but a recent work showed that the Cry11Aa toxin also associates with the epithelium from anterior and medium midgut regions, indicating that other molecules could be involved in this interaction [35]. After intoxication with Bti toxins, some histopathological effects such as severe vacuolization of the cytoplasm, microvilli damage, columnar cell fragmentation, massive degradation of the caeca gut structure, and cell lysis were observed [35,72,73,74].

#### 2.1.2. Cyt1Aa Toxin

The Cyt1Aa toxin has a single α-β domain that contains two α-helix surrounding a β-sheet [75]. This toxin interacts directly with phospholipids from the midgut cells; therefore, its action is independent of the presence of specific protein receptors [36,76,77]. The localization pattern of the Cyt1Aa toxin on the cell microvilli along the whole larvae midgut shown by recent studies corroborates its unspecific binding to the cell membrane [35,78]. Two models of action were proposed for Cyt1Aa. The first is the pore formation model that consists of cation-selective channel formation after toxin oligomerization, leading to cell lysis and osmotic shock [79,80,81]. In this model, the two outer α-helices layers of the Cyt1Aa move and expose the β-sheet structure, allowing the insertion of the β-barrel region into the cell membrane to form the pore [75,82,83]. It has been shown that the N-terminal region is responsible for the toxin oligomerization, and the C-terminal region is involved in the binding of the toxin to the membrane [84]. Specific amino acid residues and protein regions that affect Cyt1Aa binding, oligomerization, and membrane insertion have been investigated [75,85,86,87]. The second model is the detergent-like model, where it is proposed that the Cyt toxin aggregates nonspecifically on the cell membrane, leading to lipid bilayer disassembly and cell death [77,88]. The mode of action of the Cyt protein could be different for distinct target membranes since it was observed that oligomerization is a key step for Cyt toxicity in *Ae. aegypti* larvae but not for red blood cells [85]. Therefore, Cyt toxin insertion by the pore formation model could occur in microvilli membranes, while a detergent membrane interaction seems to be related to its hemolytic activity [87].

Cyt1Aa is a versatile toxin that can act alone or in synergy with Cry toxins. Nonetheless, this toxin presents low individual toxicity to mosquitoes, and its more important participation in toxicity of Bti seems to be related to its role as a receptor for Cry toxins since the larvicidal effect provided by the combined action of Cyt and Cry toxins is considerably higher than that of the toxins alone [36]. Recently, the activation of Cyt1Aa was studied through serial femtosecond crystallography analysis [89], showing that Cyt can aggregate on the membrane bilayer and form large pores with a great number of monomers being detected. These aggregates of the Cyt toxin on the membrane could function as a Cry toxin receptor, inducing the synergistic effect of these proteins. Cyt1Aa is also involved in the synergy with the Bin toxin from *L. sphaericus* [78].

#### 2.1.3. Synergistic Interaction of Cyt1Aa with Cry and Bin Toxins

The synergy of the Cry and Cyt toxins from Bti was first observed using in vivo assays analyzing the insecticidal activity against mosquito larvae [90,91,92]. The higher toxicity of the whole Bti crystal, compared to the activity of the individual toxins, could be explained by a synergistic effect of the Cyt and Cry toxins [31]. The molecular basis of this synergy involves the role of the Cyt1Aa toxin as a surrogate receptor for the Cry toxins inducing their oligomerization (Figure 2) and binding to the microvilli membrane [12,93]. The Cyt1Aa toxin is likely to be the most important factor behind the lack of resistance to the whole Bti crystal. The synergy mechanism of the Cry and Cyt toxins depends on their binding interaction. The specific binding epitopes on Cyt1Aa, Cry4Ba, and Cry11Aa responsible for this interaction were identified, and mutations in such sites affected the synergy without affecting their individual toxicity against *Ae. aegypti* larvae [47,94,95,96]. After the binding of Cyt1Aa to the midgut membrane, this protein interacts with Cry11Aa inducing its oligomerization [47,94]. Another study showed that the oligomerization of Cyt1Aa is necessary for its individual toxicity but not for the synergy with Cry11Aa against *Ae. aegypti* larvae since Cyt1Aa mutants affected in oligomerization were still able to synergize with Cry11Aa [86]. The in vivo localization of the Cry11Aa and Cyt1Aa toxins during their synergistic interaction was analyzed at a nanoscale resolution [35]. These proteins showed an ordered array in the microvilli, where Cry11Aa was found below Cyt1Aa, facing the cell cytoplasm. This interaction depends on Cry11Aa toxin oligomerization since the non-toxic mutant Cry11Aa-E97A, affected in its oligomerization, showed an inverted array when tested with Cyt1Aa. This dynamic organization pattern in the cell microvilli is consistent with the model of Cyt1Aa acting as a receptor of Cry11Aa [35]. It was also observed that Cyt1Aa can interact with other Cry toxins such as Cry2Aa, which is naturally active against lepidoptera, resulting in a slightly higher toxicity against *Cx. p. quinquefasciatus* larvae [97]. Other studies have also shown synergy between Cyt1Aa, Cry4Aa, Cry4Ba, and Cry11Aa against *Simulium* spp. [98]; Cyt1Aa and Cry10Aa against *Ae. aegypti* [99]; Cyt1Aa with Cry4Ba and Cry11Aa against *An. albimanus* [100]; and Cyt2Ba and Cry10Aa against *Ae. aegypti* [28].

Another important feature of Cyt1Aa is its synergy with unrelated toxins such as the Bin toxin from *L. sphaericus*. This protein is a heterodimer composed of BinA and BinB proteins and shows high toxicity against mosquito larvae such as *Culex* and *Anopheles*, which have specific receptors for the BinB component in the midgut microvilli (see Section 2.2.1). The Bin toxin is not active against *Ae. aegypti* larvae, and this refractoriness is due to the lack of such receptors [101,102]. In vivo synergy of Cyt1Aa with the Bin toxin was observed against *Ae. aegypti* and Bin-resistant *Cx. p. quinquefasciatus* larvae, whose midgut epithelium lack receptors for the Bin toxin [103,104,105]. It was suggested that Cyt1Aa enables the internalization of Bin on resistant larvae. Recently, the analysis of the molecular mechanism of this synergy showed that it is not based on a specific interaction between the Bin and Cyt toxins. It was demonstrated that the BinA toxin was internalized on the midgut cells in the presence of Cyt1Aa, but not in the presence of a mutant Cyt1AaV122E affected in its oligomerization and pore formation activity, suggesting that the pore formation activity of Cyt1Aa facilitates the transport of BinA into the midgut cells allowing its toxic intracellular effect [78]. The large pores formed by Cyt1Aa, observed by Tetreau et al. [89], could explain how molecules, such as the BinA subunit, could be internalized into the midgut cells. Therefore, Cyt is an important toxin that can improve the toxicity of other toxins by distinct mechanisms resulting in high synergistic effects.

### 2.2. Lysinibacillus sphaericus

*L. sphaericus* strains have been initially classified according to their mosquitocidal activity as low, moderate, or highly toxic strains [13]. The most toxic strains are characterized by the production of the crystal that contains the binary (Bin) protoxin [106]. Bin is a heterodimeric protein composed of two subunits, BinA (42 kDa) and BinB (51 kDa). None of them has individual activity, but they can act in synergy in equimolar concentrations, as they are found in the crystals produced by the highly toxic strains [107,108]. Although other mosquitocidal toxins can also be produced by *L. sphaericus* [14] (see Section 2.3), the Bin crystal is the main active ingredient in the commercial products available to date, which are based on highly toxic strains such as 1593, 2362, and C3-41 [5,109]. The decoding of the *L. sphaericus* genome enabled a better understanding of the evolution of toxins produced by the different strains and their association with the virulence phenotype [110,111,112,113,114]. A comparative analysis of genomes from high, moderate, low or non-toxic strains, revealed that the highly toxic strains exhibit strong syntenic relationships and share a “chromosome backbone” from a common ancestor, where the number of predicted genes ranged from ~4470 to 4701 [24,110]. The *bin* toxin genes, which are present only in a subset of toxic strains of *L. sphaericus*, are highly conserved showing high identity levels among the different serotypes and isolates [112].

The mode of action of crystals containing the Bin protoxin shows similar initial steps as those described for Bti: crystal ingestion by larvae and the solubilization of crystals under the midgut alkaline pH condition to release the protoxin that is converted into active toxin after proteolytic processing [115,116]. Regarding the interaction with the midgut epithelial cells, the action of the *L. sphaericus* Bin toxin relies on a high-affinity binding interaction with a single class of receptors. This last step is completely different from the complex interaction of the Bti toxins with several midgut receptors [27]. A major feature of the Bin toxin is its potent and specific larvicidal action combined with excellent persistence under field conditions. However, the mode of action depending on the interaction of the toxin with a single receptor protein can be disrupted, generating high levels of resistance. One important aspect is that the findings on the mode of action of toxins from Bti and Bin crystals showed that they can be used together to overcome resistance.

#### 2.2.1. Binary Toxin

The *binA* and *binB* genes of 1113 bp and 1347 bp encode the BinA and BinB proteins of 370 and 448 amino acids, respectively, whose sequences display 28% identity and 46% similarity, suggesting a common origin [13,117]. These two proteins are translated from a single mRNA regulated by a promoter located upstream of the *binB* gene, whose transcription starts prior to the end of the bacterial exponential growth and continues during the stationary phase of growth [118]. The arrangement of the *bin* toxin genes cluster in the chromosomal contig is conserved in several *L. sphaericus* strains [110,112,119,120]. The binary protoxin (Bin), produced in the form of crystalline inclusions, was initially classified according to four types (Bin1, Bin2, Bin3, and Bin4) based on partial DNA sequence of the *bin* genes [120]. The Bin1 toxin is found, for instance, in the IAB59 strain, while Bin2 was found in the 2362 and 1593 strains, with both proteins being highly toxic and showing high binding capacity to midgut microvilli of *Cx. p. quinquefasciatus* larvae [121]. Most studies on the mode of action of the Bin toxin have been analyzed with the Bin1 and Bin2 proteins. Recently, the Bin toxin was classified as a “Beta sheet toxin”, according to its structure and was grouped in the “Toxin_10 family” [122]. All the proteins from this family act with their partner proteins to form Binary toxins as the homologous BinA and BinB molecules [14]. Early studies of the functional domains of bin subunits revealed that the receptor binding function is performed by the BinB component, whereas binA is responsible for the toxic activity inside the cell. The optimal toxicity is achieved at an equimolar concentration of the subunits [40,108,123]. BinA and BinB are monomeric proteins, either as protoxins or as activated toxins. When activated, they combine and form a heterodimer [124]. These toxins have two domains: a trefoil domain and a pore formation domain [117], and no evidence of oligomer formation was detected for their toxic action [124,125,126], contrary to the oligomerization that has already been demonstrated for the Cry toxins [46].

The C-terminal domain of the BinA component (42 kDa) is associated with cell toxicity [127] and might also be involved in the ability to form pores in the intestinal epithelium, supporting the internalization of the toxin [39,128]. Some specific residues in this subunit have been already identified as necessary for BinA toxicity [82,129,130]. The N- and C-terminal domains play an important role for the BinA–BinB interaction [131,132,133] that is needed to promote binding of BinB to the cell receptors and BinA’s entry into the cells.

The N-terminal region of the BinB subunit (51 kDa) is responsible for the interaction with its receptor located in the intestinal epithelium, and within this segment some residues are critical for this interaction [41,42,134,135]. Like BinA, the structure of the BinB subunit has a predominance of β-sheets [117]. The N-terminal domain has two cysteine residues that are required for toxicity [136]. The C-terminal region of BinB participates in the interaction with the BinA component [39,40,42]. This C-terminal domain has a cluster of aromatic residues, which are critical for the proper conformation of toxins and insertion into the membranes [137]. The resolution of the BinA-BinB crystal structure revealed important molecular events in the toxin’s life cycle that involve structural rearrangements of the protein triggered by alkaline conditions and proteolytic cleavages [117,138]. These changes include the detection of pH switches that facilitate the solubilization of the crystal, a heterodimeric interface that remains bound after dissolution, carbohydrate binding modules in BinA that can direct heterodimers to the cell surface, and a proteolytic maturation that triggers heterodimer dissociation and remodeling [117].

#### 2.2.2. Bin Toxin Interaction with Cell Receptors and Intracellular Action

The action of the Bin protoxin has been mostly studied in insect species belonging to the *Cx. pipiens* complex. After protoxin processing, the activated toxin recognizes and binds to specific receptors located on the midgut epithelium of the larvae [102]. In the most susceptible species of *Cx. pipiens*, the binding of the Bin toxin is regionalized in the gastric caeca and posterior midgut (Figure 3A), while for some *Anopheles* larvae, which are less susceptible than *Cx. pipiens*, the binding pattern in the gut is less defined [40,139,140]. The binding affinity of Bin to the larvae midgut directly correlated with the in vivo susceptibility of the species [102,121,141,142,143,144,145,146]. In *Ae. aegypti* larvae, which is refractory, the Bin toxin binding to the midgut cannot be detected (Figure 3B).

The receptors of the Bin toxin, characterized in three major target species, are ortholog midgut-bound α-glucosidases that were denominated Cpm1 for *Cx. p. pipiens* and correspond to maltase 1 [147,148], Cqm1 for *Cx. p. quinquefasciatus* corresponding to maltase 1 [149], and Agm3 for *An. gambiae* corresponding to maltase 3 [150]. *Ae. aegypti* has a gene that encodes an ortholog, Aam1 (corresponding to maltase 1), with 74% identity shared with Cqm1. Aam1 is also expressed as a membrane protein in the midgut epithelial cells, but this protein is not able to bind to the Bin toxin, which is the reason for the larvae refractoriness [101]. Cpm1 was the first receptor characterized showing 97% and 66% identity with Cqm1 and Agm3, respectively. These α-glucosidases (EC 3.2.1.20) belong to the large family of α-amylases proteins that have the ability to hydrolyze α-1-4 links between glucose residues of carbohydrates [151]. They display four α-glucosidases conserved domains and the (α-ß)_8_ barrel fold for the glycoside hydrolases (GH) from the GH-13 family, which comprises most mosquito α-glucosidases [152]. The α-glucosidases from mosquito larvae have been poorly characterized [153]. However, the catalytic activity of the native or recombinant Cqm1 was demonstrated, indicating its potential ability to participate in carbohydrate digestion [154,155,156].

The *cpm1*, *cqm1*, and *agm3* genes encode proteins of 580 to 588 amino acids that display the four conserved α-glucosidase domains, showing predicted glycosylation sites and a signaling sequence for a GPI-anchor at the C-terminal end [147,149,150]. Their expression as midgut membrane-bound proteins is essential for the binding to the Bin toxin, and gene mutations that disrupt their expression as GPI-anchored proteins have been recognized as the most important mechanism that confers resistance to the Bin toxin in mosquitoes (see Section 4). The expression of Cqm1 recombinant proteins in some cell lines has been used to demonstrate its capacity to bind to the bin toxin, to mediate the cytopathological effects, and to assess its catalytic activity [154,155,156,157,158]. Functional assays using such recombinant proteins showed that the N-terminal region of Cqm1 is required for its binding to the Bin toxin [154]. The X-ray crystallographic analysis of Cqm1 revealed three structural domains [159]. The residues from the domain B adopt the (α-ß)_8_ barrel fold and the region implicated in receptor binding was located in the loops of domain A, including also some residues of domain B [160]. Folding analysis indicated that Cqm1 is found as a stable dimer anchored in the apical membrane of the midgut cells [156].

Post-binding events are still under investigation, and it was shown that in *Cx. p. quinquefasciatus* the BinB subunit binds to the receptor and the BinA subunit is found inside the midgut cells (Figure 3A). The most commonly observed pathological alterations reported in the midgut epithelial cells of Bin-treated larvae were the destruction of microvilli, mitochondrial swelling and damage to the inner membrane, intense cytoplasmic vacuolization, and breakdown of endoplasmic reticula [161,162,163,164,165]. Damage in the muscular and neural tissues of the larvae was also reported [164]. The localization studies of Bin subunits in the treated *Cx. p. quinquefasciatus* larvae showed that BinB binding to the Cqm1 receptor is a step that is required for the internalization of both BinA and BinB subunits, which could occur by endocytosis [163,166]. These studies have shown that toxicity is directly associated with the presence of the BinA subunit inside the cells, which depends on the interaction of BinB with the receptor [123,166]. However, in cells deprived of Bin receptors, such as Bin-resistant *Cx. p. quinquefasciatus* and naturally refractory *Ae. aegypti* larvae, the entry of BinA can be mediated by the Cyt1Aa toxin (Figure 3C) and is associated with increased larvicidal activity [78,167]. The high toxicity of a chimeric BinA-Cyt1Aa toxin [168] or pegylated-BinA [169] was also reported. Cyt1Aa has the capacity to induce entry of Bin toxin into the cell, which is due to the ability of Cyt1Aa to form pores in the apical membrane [78]. Therefore, the internalization of BinA into the midgut cells, either by the interaction of BinB with the cell receptor, or by an alternative mechanism, is essential to cause injury and larval death.

Both Bin subunits were found to display the capacity to form pores in culture cells or artificial membranes [128,134,170,171,172]. Madin-Darby canine kidney cells expressing the Cqm1 receptor on the membrane also showed that Bin, after binding to the receptor, had the ability to form pores and to induce autophagy [173], which is consistent with cytoplasmatic vacuolization, one of the most prominent alterations resulting from Bin intoxication [161,162]. The activation of the intrinsic apoptosis pathway by Bin action has been also investigated, as mitochondria are a major intracellular target of the Bin toxin [165]. A transcriptome analysis comparing untreated and Bin-treated *Cx. p. quinquefasciatus* larvae revealed differential expression of transcripts involved in mitochondria mediated apoptosis and autophagy responses [174]. Another study comparing susceptible and Bin-resistant larvae revealed an outstanding differential expression of transcripts involved in apoptosis and DNA metabolism [175]. These data suggest that both apoptosis and induced autophagy mechanisms could be involved in the larval death caused by the Bin toxin. It has been also proposed that the intracellular action of BinA could be associated with its ability to bind to N-glycosylated proteins [176].

### 2.3. Other Toxins

Other toxins produced by *L. sphaericus* and Bt strains have been studied but not yet used in the development of commercial products. In addition to Bti, other Bt strains can produce mosquitocidal toxins, and they were classified into three groups [177,178]. The Class 1 strains appear to be the highly similar to Bti [178]. This is the case for the *B. thuringiensis* svar. *morrisoni* (serotype 8a:8b) PG-14 strain, which showed high and selective toxicity against *Ae. aegypti* and *Culex molestus* [179,180]. The crystals from this strain include protoxins immunologically related to those of Bti, including Cry4A, Cry4B, Cry10A, Cry11A, Cry1Ac, and Cyt1Aa2 [181,182,183]. The Class 2 strains contain multiple proteins different from the proteins found in Bti crystals [178], and the most studied strains are *B. thuringiensis* svar. *jegathesan* and *B. thuringiensis* svar. *medellin*. To date, eight protoxins (Cry11Ba, Cry19Aa, Cry24Aa, Cry25Aa, Cry30Ca, Cry60Aa, Cry60Ba, and Cyt2Bb) have been identified in *B. thuringiensis* svar. *jegathesan* [184], and they can be as toxic as Bti to *Anopheles stephensi*, *Ae. aegypti,* and *Cx. pipiens* larvae [177,185]. Two strains of *B. thuringiensis* svar. *medellin* have been characterized [186,187], and one of them showed high toxicity comparable to Bti, but the crystal contains different polypeptides including Cry11Bb, Cry29A, Cry30A, CytlAb, and Cyt2Bc [183,188,189,190]. Cry11Bb is the most toxic component with an activity comparable to Cry11Ba [191,192], while no mosquitocidal activity was reported for Cry29A or Cry30A [193]. CytlAb is as hemolytic as CytlAa, but less active against mosquitoes [190]. Cyt2Bc also has mosquitocidal activity against *Ae. aegypti*, *An. stephensi*, and *Cx. p. quinquefasciatus*, including larvae resistant to the Bin toxin [188]. Class 3 includes strains that produce polypeptides different from those found in Bti but that show low toxicity against mosquito larvae [178]. This group includes some strains with high activity against other insect orders such as *B. thuringiensis* svar. *kurstaki* (serotype HD-1), which is the most commonly used for controlling lepidopteran larvae. This strain can produce the Cry2Aa toxin, which has a dual specificity against dipteran and lepidopteran larvae [194]. Other examples of strains from this class are *B. thuringiensis* svar. *kyushuensis* [195], *B. thuringiensis* svar. *darmstadiensis* [196], *B. thuringiensis* svar. *fukuokaensis* [197], *B. thuringiensis* svar. *galleriae* [198], *B. thuringiensis* svar. *higo* [199], and *B. thuringiensis* svar. *aizawai* [200].

In addition to the Bin toxin, four insecticidal toxins were found in *L. sphaericus* strains: mosquitocidal toxins (Mtx), sphaericolysin, S-layer proteins, and Cry48Aa/Cry49Aa [14,201,202]. The production of Mtx-toxins was identified during the bacterial vegetative stage, and it was shown that they display low activity because they are subjected to proteolytical degradation [203,204,205]. In contrast, the Mtxs expressed as recombinant proteins in *Escherichia coli* display high activity against dipteran larvae [206,207]. The mixture of recombinant Mtx and Binary toxins can also display an increased activity and be useful for managing Bin resistance [208,209]. Sphaericolysin (53 kDa) is a cytolysin whose insecticidal activity was observed when injected into *Blatella germanica* and *Spodoptera litura*. However, no action against dipterans was reported [210]. The S-layer proteins (120-130 kDa) found associated with vegetative cells and spores of some *L. sphaericus* strains (e.g., 2362 and C7), can contribute to the larvicidal activity against *Cx. p. quinquefasciatus* [201,202,211]. In addition to these, another promising active ingredient are crystals containing a Binary protoxin composed of Cry48Aa/Cry49a toxins, which are produced by some Bt strains such as IAB59 [212]. This is also a two-component toxin formed by Cry48Aa (135 kDa), a typical three-domain structure toxin from the Cry toxins family, and Cry49Aa (53 kDa), which has a similarity to other Cry Binary toxins [14,212,213] and has been recently named Tpp49 [16]. The optimal larvicidal activity is only achieved in the presence of an equimolar concentration of the Cry48Aa and Cry49a subunits [212,214]. However, the production of Cry48Aa in native strains is low and possibly unstable [212]. If the expression of Cry48Aa/Cry49Aa is optimized in recombinant bacteria and toxins are administrated in an equimolar concentration, they display high larvicidal activity similar to the Bin toxin [214]. The spectrum of Cry48Aa/Cry49Aa action seems to be restricted to *Cx. p. quinquefasciatus* based on a bioassay screening that included other dipterans species [214]. Some steps of the mode of action of Cry48Aa/Cry49Aa are similar to Bin and Bti protoxins [212,214,215,216,217], and molecules such as APNs, ALPs, and maltases, in addition to other proteins, were identified as the toxin ligands in *Cx. p. quinquefasciatus* larvae [217]. Cry48Aa/Cry49Aa could be considered an important alternative for mosquito control due to its action against *Cx. p. quinquefasciatus* larvae that are resistant to the Bin toxin [212,215]. The continuous search for novel mosquitocidal toxins with a high and strategic mode of action is essential for the development of microbial-based products with improved characteristics [122,218].

## 3. Applications for Mosquito Control

Microbial larvicides based on the insecticidal crystals of Bti and *L. sphaericus* have been used for mosquito control since the 1980s [3,5,109]. Bti has been employed to fight mosquito and black-flies and, even after decades of widespread use, field resistance to Bti crystals has not been documented (see Section 4). On the other hand, Bti crystals are vulnerable to abiotic (e.g., photolysis) and biotic factors (e.g., high content of organic matter) that reduce their residual effect in mosquito habitats [5,219,220,221,222]. *L. sphaericus*-based larvicides have been mostly used to control *Cx. pipiens* and *Anopheles* displaying advantages such as persistence in water polluted with organic materials and the ability to be recycled in the cadavers of the mosquito larvae [5,223,224]. However, the use of *L. sphaericus* larvicides as the single control agent can lead to the resistance of the mosquito larvae to the Bin toxin (see Section 4). It is important to highlight that, in the past Bti and *L. sphaericus* larvicides were used as single control tools in mosquitoes or black-flies control programs that showed effectiveness in a range of scenarios while, nowadays, they are used as part of integrated measures [225]. Here, we show some examples of applications of Bti and/or *L. sphaericus* larvicides, considering their use within a scenario of integrated mosquito control in recent trials (Table 1).

The commercial utilization of Bti took place very soon after its discovery (1977–1982), being a remarkable example of a successful biotechnological development [226]. Bti was first used to fight *Simulium* spp. in the outstanding Onchocerciasis Control Program carried out in West Africa in 1982 in order to replace organophosphate larvicides that were used until then [227,228,229]. A program to control the floodwater mosquito, *Aedes vexans,* a nuisance pest across a wide area of the Rhine Valley in Germany, was carried out over more than four decades by the German Mosquito Control Association-KABS [3,18,230]. Since its introduction, Bti has also been a key for overcoming the resistance that was developed by *Simulium* and *Aedes* populations to organophosphates [231,232,233,234,235] and to prevent the establishment of invasive species, such as *Aedes albopictus, Aedes japonicus*, and *Aedes koreicus* [236,237,238,239,240,241,242,243,244,245]; it is also a safe control agent for reducing mosquito proliferation in environmentally protected areas [246,247,248,249]. More recently, Bti larvicides have been employed to control other species and used in combination with other control approaches. For instance, Bti has been used for *Anopheles* control in association with “Long Lasting Insecticide Treated Nets” (LLINS) and “Indoor Residual Spraying” (IRS) [250,251,252,253,254,255,256,257]. The control of mosquito larvae in the breeding sites located close to houses in malaria-endemic areas has been highly effective in reducing adult reproduction and disease transmission, as shown by trials performed in sub-Saharan Africa [4]. The innovative use of Bti includes novel approaches such as its association with lethal ovitraps to prevent *Aedes* larvae development [258,259,260], its use in “Attractive Toxic Sugar Baits” (ATSB) and sugar patches to target adults [261,262,263], and its use in spatial spraying to reach cryptic breeding sites [264,265,266,267]. The use of Bti combined with *L. sphaericus* is also of crucial importance for the management of *Cx. pipiens* resistance to *L. sphaericus*, as will be discussed below.

The isolation of *L. sphaericus* strains highly toxic against mosquitoes producing crystals with the Bin toxin (e.g., 1593, 2362, and C3-41) induced its commercial utilization in several countries [5,8]. *L. sphaericus* was first introduced to control *Cx. p. pipiens* that were a nuisance pest in the south of France in 1987 [189]. Soon after, the WHO supported field trials in some countries endemic for filariasis to evaluate its effectiveness in the control of *Cx. p. quinquefasciatus* that acted as the main vector in urban areas characterized by poor sanitation and high mosquito proliferation [18,109,268]. Other applications for vector control include its use in India against *An. stephensi*, in China against *Anopheles sinensis*, and in Brazil against *Anopheles darlingi* [189,269,270,271,272]. Therefore, *L. sphaericus* larvicides have been used for controlling *Culex, Anopheles,* and other genera in urban or rural areas from several countries showing outstanding field performances [273,274,275,276,277,278,279,280,281,282,283].

The use of *L. sphaericus* larvicides can lead to development of mosquito larvae resistance as reported in some field-treated populations of *Cx. p. pipiens* [26,27]. Nevertheless, studies aiming to characterize Bin resistance demonstrated that Bti crystals were still active against these Bin-resistant larvae, as Bti toxins targets different receptors in the midgut epithelial cells (see Section 2). Given this scenario, approaches to manage or delay Bin resistance based on the association of *L. sphaericus* and Bti crystals have been developed (Table 1). The combination of their active ingredients can offer advantages such as an enhanced spectrum of action, longer persistence, and a lower risk of resistance selection [246,281,284,285,286]. The treatment of mosquito breeding sites with Bti and *L. sphaericus* larvicides in rotation, integrated or not with LLINs, has been used to reduce the density of anophelines and to improve malaria control in Africa [252,254,287,288,289,290]. Bti and *L. sphaericus* larvicides used in rotation along with environmental management practices were adopted to control *Cx. p. quinquefasciatus* in São Paulo city, Brazil, without issues of resistance selection [146,291]. In addition, these larvicides have been mixed and applied together [292,293].

The successful experiences of using Bti in combination with *L. sphaericus* led to the development of commercial combined products containing crystals of both bacteria. Some of them are long-lasting microbial larvicides whose formulations provide a slow release of the active ingredients over 90 to 180 days [294,295,296], and they have been used to control mosquito larvae in a variety of landscapes and purposes (Table 1). In urban areas, such larvicides have been used in several countries, such as in Italy, Switzerland, and Spain to control *Ae. albopictus* [237,239,297]; in Netherlands and USA against *Ae. japonicus* [240,284]; in the USA against *Cx. pipiens* and *Culex restuans* [240,284,298,299]; in Colombia and Brazil to control *Cx. p. quinquefasciatus* and *Ae. aegypti* [281,300]; in Kenya against *Cx. p. quinquefasciatus* and *An. gambiae* [301]; and in Senegal against *Anopheles arabiensis* [302]. These combined larvicides are viable options for controlling mosquito populations and interrupting disease transmission, along with other measures [4,294,295,296,303,304].

**Table 1 toxins-13-00523-t001:** Field trials using *Bacillus thuringiensis* svar. *israelensis-* (Bti) and *Lysinibacillus sphaericus*-based larvicides used for mosquito control in rotation, as a mixture or as combined products.

Larvicide-Scheme	Control Intervention ^(a)^	Country	Target Species	Scenario	Outcome	Reference
*L. sphaericus* and Bti in rotation	Larvicides	Kenya	*Anopheles gambiae* *Anopheles funestus*	Rural	Reduction of larval density and human biting exposure	[287]
		Gambia	*An. gambiae*	Rural	Reduction of pupal and larval densities	[288]
		Tanzania	*An. gambiae* *Culex quinquefasciatus*	Urban	Reduction of larval abundance and malaria transmission	[289]
		Cote d’Ivoire	*An. gambiae* *An. funestus* *Culex* spp.	Urban	Reduction of breeding sites number and biting rates	[290]
	Larvicides, ITN	Kenya	*An. gambiae* *An. funestus* *Anopheles arabiensis*	Urban	Reduction of larval density and new malaria infections	[252]
	Larvicides, ITN, and other measures	Tanzania	*An. gambiae* *An. funestus* *Cx. quinquefasciatus*	Urban	Reduction of malaria infections	[253,254]
	Larvicides and environmental management	Brazil	*Cx. quinquefasciatus*	Urban	Reduction of mosquito density	[146,291]
*L. sphaericus* and Bti in mixture	Larvicides	Turkey	*Culex pipiens*	Urban	Reduction of larval density	[292]
*L. sphaericus*/Bti-combined in a single product	Larvicides	USA	*Culex tarsalis,* *Aedes melanimon*	Sylvatic	Reduction of larval and pupal density	[246]
		Kenya	*An. gambiae*	Rural	Reduction of pupal density, and indoor- and outdoor-biting mosquitoes	[294]
		Kenya	*An. gambiae* *An. funestus*	Rural	Reduction of larval density	[295]
		Kenya	*An. gambiae* *Cx. quinquefasciatus*	Urban/peri-urban	Reduction larval density	[301]
		Brazil	*Cx. quinquefasciatus Aedes aegypti*	Urban	Reduction of larval density	[281,305]
		Spain	*Aedes albopictus*	Urban/indoor catch basins	Reduction of mosquito emergence	[237]
		Switzerland	*Ae. albopictus*	Urban	Entomological data not available	[239]
	Larvicides, ITN, and IRS	Kenya	*An. gambiae* *An. funestus* *An. arabiensis*	Rural	This field trial is ongoing	[296,303]
	Larvicides, and other measures	Italy	*Ae. albopictus*	Urban	Reduction off egg density	[297]
	Larvicides and source reduction	Netherlands	*Aedes japonicus*	Peri-urban/allotment garden	Reduction of larval abundance	[240]
*L.**sphaericus*/Bti-combined, Bti, and Methoprene	Multi-larvicides	Senegal	*An. arabiensis*	Urban	Reduction of larval density	[302]
*L.**sphaericus*/Bti-combined, L. sphaericus, Bti and Spinosad		USA	*Cx. pipiens* *Culex restuans* *Ae. japonicus*	Urban	Reduction of immatures	[284]
*L.**sphaericus*/Bti-combined and Triflumuron		Colombia	*Cx. quinquefasciatus Ae. aegypti*	Urban	Reduction of immatures	[300]
*L.**sphaericus*/Bti-combined and Spinosad		USA	*Cx. pipiens*	Urban	Reduction of larval density	[298]
*L.**sphaericus*/Bti-combined, *L. sphaericus* and Spinosad		USA	*Cx. pippiens*	Urban	Reduction of pupae production	[299]
*L.**sphaericus*/ Bti-combined and *L. sphaericus*		Brazil	*Anopheles darlingi*	Rural/fish farming ponds	Reduction of larval density	[306]

(a) ITN: insecticide treated net; IRS: insecticide residual spray.

## 4. Resistance Issues

Resistance and safety of larvicidal compounds to mosquito control are prominent issues and they need to be continuously assessed. This section summarizes results from studies that have investigated Bti resistance and also the reports of *L. sphaericus* resistance, which was already detected. The environmental and human safety issues of the major insecticidal toxins produced by Bti and *L. sphaericus* have been studied since the early characterization of these entomopathogenic bacteria [307,308,309,310,311] and multiple reports have been published since then. Detailed environmental assessments have been conducted regarding to Bti applications for several decades under the light of actual regulation for the use of biocides in Europe [234,312]. In this scope, Bruhl et al. (2020) published a complete review focused to the description of Bti persistence and its environmental impact, including direct effects on the non-target organisms and indirect effects related to the food-chain. The authors presented a detailed analysis and highlight caution regarding to the use of Bti in environmental protected areas, as well as the need of improved monitoring strategies of such effects and adoption of alternative control measures for such habitats. Among some critical issues, we can mention the persistence of Bti spores in the soil and its potential impact in microbiota. A recent study analyzed the possible impact of multiple Bti applications in the soil of Riparian wetlands of Switzerland on the population of *Bacillus cereus*, but no direct correlation was found [313]. In terms of safety to other organisms, studies assessing the Bti impact on chironomides, a key element in the food-web chain, showed that the actual criteria of the biocide regulation used in Europe could be underestimated [314,315]. It is worth noting that, in some scenarios, chironomides can also be a target species. Some initial assessments of the combined set of Bti and *L. sphaericus* crystals on non-targets organisms have also been investigated [316,317]. To date, Bin crystals and Bti crystals are still considered as safe compounds that effectively control several dipteran species larvae of medical importance. However, improved safety assessments should be continuously performed to deep our knowledge about their potential ecological implications, in particular, focusing their use in environmental-sensitive areas.

### 4.1. Resistance to Bti

To date, there are no reports of insect field resistance to Bti, although Bti based products have been used in multiple mosquito control programs since 1982 [3,227,230,318,319]. The synergistic mode of action of the insecticidal protoxins from Bti crystal is considered a key factor underlying the lack of resistance development. Assessments of larvae susceptibility to Bti crystals from several Bti-treated mosquito populations worldwide have shown lack of resistance to the whole crystal, which is the active ingredient of the available larvicides (Table 2). The control program for *Ae. vexans* in the Rhine Valley in Germany is an example of the long-term utilization of Bti-larvicides without resistance issues [3,319]. The assessment of the Bti susceptibility of non-treated populations has shown a range of natural variations before the introduction of this microbial larvicide (Table 2). Variations in the resistance ratios (RR) ranged from 0.8- to 8-fold for *Aedes* species [231,232,234,320,321,322,323,324,325,326,327,328], from 1.5 to 12.5-fold for *Cx. p. pipiens* [234,329,330], and from 0.8- to 5.9-fold for *Anopheles* [234,303,331]. The range of variations found among the treated populations was similar to those observed in non-treated samples from the analyzed species (Table 2), reinforcing the lack of resistance development to Bti crystals [232,322,332,333,334,335,336,337]. It is worth noting that two *Cx. p. pipiens* populations in New York State that showed RRs of 14- and 41-fold are an exception in this scenario [338], and there was no evidence that the resistance ratios found were a consequence of Bti treatments.

Attempts to select insect resistance to the whole Bti crystal under laboratory conditions have also failed as RR values were less than three-fold (Table 3), which are not biologically meaningful, considering the range of variations of Bti susceptibility recorded for non-treated populations (Table 2) [339,340,341,342,343,344]. On the other hand, laboratory selection of resistant populations to a single Bti toxin were reported [53,345,346,347], which is an expected consequence since the synergy of the whole set of toxins is lost under such conditions. It is also worth noting that larvae selected using whole Bti crystals do not display resistance to the Bti crystal; however, such larvae can display a reduction in susceptibility to some single Cry toxins, suggesting that monitoring the susceptibility to individual Cry toxins could be a marker for the analysis of populations subjected to chronic Bti exposure (Table 3). This was the case in a laboratory colony that was selected for Bti crystals, which still showed susceptibility to Bti crystals but displayed resistance to Cry4Aa (68-fold), Cry4Ba, and Cry11Aa (9-fold) toxins [344]. However, another study that evaluated a colony selected for 30 generations with Bti crystal treatment showed that the larvae were still susceptible to Bti crystals and also to Cry11Aa and Cry4Ba toxins, indicating that reduction in susceptibility to individual toxins might not necessarily occur under chronic Bti exposure [339]. Although Bti displays a low potential for resistance development, the analysis of receptor expression, proteolytic processing of toxins, immune response, and other pathways are important factors to be further investigated in order to increase our knowledge on the mode of action of these proteins [348,349]. The potential impact of Bti exposure on the life traits of mosquitoes has also been studied (see Section 5).

### 4.2. Resistance to L. sphaericus (Bin Toxin)

The greatest challenge related to the long-term use of *L. sphaericus* larvicides is the emergence of insect resistance to the Bin toxin. The selection of resistant insects depends on general factors, such as the use of larvicides for long periods of time that increases the selection pressure as well as on specific factors such as the mode of action of the Bin toxin itself [8,25,27]. Resistance to the Bin toxin was detected in *Cx. p. pipiens* and *Cx. p. quinquefasciatus* field treated populations and in laboratory-selected colonies, as summarized in Table 4. The first record was a *Cx. p. pipiens* population from France that was subjected to five years of treatment and showed high resistance ratio to the Bin toxin (RR > 20,000) [350,351]. Other cases of high resistance were recorded in treated populations from India, China, Thailand, Tunisia, and USA [25,143,272,351,352,353,354,355,356,357]. Selection of *Cx. p. pipiens* and *Cx. p. quinquefasciatus* under laboratory conditions using *L. sphaericus* strains also showed that high resistance could be achieved [357,358]. Two laboratory strains were selected using the IAB59 strain [358,359,360] that produces Bin and Cry48Aa/Cry49Aa crystals, and high levels of resistance to Bin were achieved, but only a moderate level of resistance was detected for the Cry48Aa/Cry49Aa toxin [217].

The resistance to Bin toxin can reach high levels because the major resistance mechanism is associated with the absence or alteration of the toxin receptor, which completely disrupts the action of this toxin on the cells [102,141,144]. The molecular characterization of the Bin resistance mechanism showed that such larvae carried alleles of the *cqm1* gene with mutations that prevent expression of the Cqm1 α-glucosidases. Normally, the Cqm1 protein is located in the midgut epithelium as a GPI-anchored protein [149,157,361,362,363,364]. A variety of missense and nonsense mutations in *cqm1* alleles that confer resistance were found, and most of them cause the production of transcripts coding for truncated proteins without the GPI anchor; therefore, they are no longer located on the cell membrane (Figure 4). The exceptions were SPHAE and TUNIS *Cx. p. pipiens* colonies, whose larvae have functional Cqm1 receptors, indicating that resistance was due to another mechanism [142,143].

The first allele conferring resistance to the Bin toxin was identified in the *Cx. p. quinquefasciatus* GEO colony (USA), which displayed a high RR of ~100,000 after laboratory selection with the 2362 strain [357]. The *cpm1_GEO_* allele exhibits a point mutation that generates a premature translation termination codon, which leads to the expression of a 568-amino acid protein without the GPI anchor [157]. The resistance of another *Cx. p. quinquefasciatus* colony (CqRL/C3-41) from China [272] was associated with the *cqm1R* allele, which is associated to one deletion that generates a truncated protein due to a premature stop codon [364]. In the resistant *Cx. p. pipiens* BP population from France, two alleles (*cpm1_BP_* and *cpm1_BP–del_*) were found [350,363]. The *cpm1_BP_* allele has a nonsense mutation that leads to the formation of a premature stop codon and synthesis of a truncated protein with 395 amino acids lacking the GPI anchor. The *cpm1_BP–__del_* allele was characterized by the insertion of a transposon, which leads to a 198 bp deletion. Such transcript encodes for a protein of 514 amino acids with the GPI anchor, but lacks 66 amino acids, and this truncated protein was unable to bind to the Bin toxin.

In *Cx. p. quinquefasciatus* from Recife city, Brazil, four *cqm1* alleles conferring resistance were detected from laboratory-selected colonies or from field samples. The resistance of the R2362 laboratory-selected colony was associated with two alleles, *cqm1_REC_* and *cqm1_REC-2_*, [149,362]. For the IAB59-selected colony, the resistance to the Bin toxin was associated with homozygous larvae for the *cqm1_REC_* allele [359,360]. This allele had a 19-nucleotide (nt) deletion, which generates a premature stop codon and a truncated protein without the GPI anchor [149]. The *cqm1_REC-2_* allele has a nonsense mutation that generates a premature translation termination codon, and the transcripts also code for a truncated soluble protein [362]. Two colonies formed by homozygous individuals for each allele (REC and REC-2) were established [362,365]. Field screenings revealed two other alleles, named *cqm1_REC-D16_* and *cqm1_REC-D25_*, which showed deletions of 16- and 25-nt, respectively, resulting in truncated transcripts [361]. DNA screenings of the *cqm1_REC_* and *cqm1_REC-2_* alleles from field populations in Recife, Brazil, without exposure to *L. sphaericus*, revealed their presence with frequencies in the order of 10^−3^ and 10^−4^, respectively [366,367]. The finding of *L. sphaericus* resistance in field populations indicated the need to adopt additional strategies to avoid the selection of such resistant alleles of the *cqm1* gene, particularly because they can provoke high levels of refractoriness [8,25,27].

On the other hand, it is important to mention that the *cqm1* resistance alleles are recessively inherited; therefore, only homozygous individuals display the resistant phenotype [141,142,144,357,359,362,364]. Another important aspect is the lack of cross-resistance to the Bin toxin by other control agents, which make them viable options to restore the susceptibility to *L. sphaericus*. Bti is a suitable candidate since Bin-resistant larvae are still highly susceptible to Bti crystals [141,358,368,369], and examples of the combination of Bti and *L. sphaericus* crystals were presented in Section 3. Other insecticidal compounds, such as Spinosad produced by the bacterium *Saccharopolyspora spinosa* [370], and insect growth regulators [300] are also compatible with *L. sphaericus*. The recombinant expression of their toxins together in *Bacilli* or other microorganisms has been demonstrated, although they have not been developed for commercial use [371,372,373,374].

**Table 2 toxins-13-00523-t002:** Susceptibility of mosquito field populations to *Bacillus thuringiensis* svar. *israelensis*.

Species	Country	No. Populations	Status ^(a)^	RR ^(b)^	Reference
*Aedes aegypti*	Malaysia	4	NT	1.4–2.0	[324]
		2	T	2–4	[334]
	Brazil	9	NT	1–1.3	[231]
		5	T	1–1.7	[231,336]
	Cameroon	4	NT	1.1–2.8	[323]
	Saudi-Arabia	1	NT	1.2	[320]
	Mayotte	1	NT	1.0	[234]
	Cape Vert	7	NT	0.8	[326]
	Martinique	1	T	1	[232]
	Laos	1	NT	0.8	[325]
	USA	1	NT	0.8–1.3	[327]
*Aedes albopictus*	Cameroon	3	NT	1.1–1.1	[323]
	Malaysia	4	NT	1.2–3.9	[324]
		4	T	1.4–1.9	[335]
	USA	2	T	≅1	[332]
	Italy	2	T	1.7	[336]
	Cameroon	5	NT	0.8–2.8	[328]
	Greece	3	NT	1.5	[321]
	China	4	NT	>5	[375]
*Aedes vexans*	Germany	3	T	≅1	[319]
		6	T	0.8–1.1	[3]
*Aedes rusticus*	France	3	NT	1.0–5.0	[322]
	France	4	T	2.8–7.9	[322]
*Culex pipiens pipiens*	Cyprus	7	NT	12.5	[330]
		10	NT	>3	[329]
	Mayotte	1	NT	1.5	[234]
	USA	31	NT	4.0	[330]
		2	T	6–33	[338]
*Cx. p. pallens*	China	1	T	6.7	[337]
		3	T	0.7–1.0	[376]
*Anopheles sinensis*	China	5	ND	1.7–5.9	[331]
*Anopheles gambiae*	Mayotte	1	NT	1.5	[234]
	Kenya	5	NT	0.8–1.5	[305]

(a) NT: nontreated population; T: treated populations; ND: not determined. (b) Resistance ratio at LC_50_ (LC for larvae from a test colony/LC for larvae from a reference colony).

**Table 3 toxins-13-00523-t003:** Selection of Culicine larvae with *Bacillus thuringiensis* svar. *israelensis* crystal or toxins under laboratory conditions that were analyzed for resistance to Bti crystal or individual toxins.

				RR ^(a)^				
Species	Country	No. Generations	Selection Agent	Bti	Cry4Aa	Cry4Ba	Cry11Aa	Reference
*Aedes aegypti*	USA	15	Bti	1.1	__	__	__	[339]
	Sri Lanka	15	Bti	1.1	__	__	__	[339]
	Brazil	15	Bti	2.0	__	__	__	[339]
		30	Bti	1.5	__	2.7	3.8	[337]
	France	18	Bti	2.0	30	14	6	[341]
		22	Bti	__	35	11	6	[340]
		30	Bti	3.5	68	9	9	[342]
	Colombia	54	Cry11Aa	__	__	__	13	[343]
	USA	27	Cry11Aa	__	66	13	124	[53]
	France	22	Cry11Aa	2.0	6	15	29	[345]
		22	Cry4Aa	1.4	65	10	5	[345]
		22	Cry4Ba	1.5	3	27	10.4	[345]
		5 ^(b)^	Bti	0.8	4.4	3.7	1.6	[345]
		33	Cry11Aa	1.7	18	36	70	[67]
		33	Cry4Aa	1.2	1018	2.7	3.4	[67]
		33	Cry4Ba	1.6	34	226	13	[67]
		14 ^(b)^	All Cry’s	1.4	14	8	5.4	[67]
*Culex* *pipiens*	USA	28	Bti	2.0	6	14	30	[338]
		28	Cry11Aa	43	__	__	__	
	India	20	Bti	2-3	__	__	__	[377]
	Egypt	20	Bti	2.8	__	__	__	[378]

(a) Resistance ratio at LC_50_ (LC_50_ for larvae from a test colony/LC_50_ for larvae from a reference colony). (b) This selected strain was a composite strain resulting from a mix of adults, in equal amounts, from each Cry selected strain (30% each at the generation 18) and 10% of adults from a susceptible Bora Bora strain.

**Table 4 toxins-13-00523-t004:** *Culex pipiens* populations or laboratory-selected colonies exposed to *Lysinibacillus sphaericus* that were investigated for resistance.

Origin	Country	Sample/Colony ^(a)^	*r* Alleles	RR ^(b)^	Inheritance ^(c)^	Binding to Receptors	Reference
Field	France	Port St-Louis	ND	>20,000	ND	ND	[349]
		SPHAE	ND	>20,000	R/S	Yes	[142,143]
		BP	*cpm1_BP_*/*cpm1_BP_-del*	>10,000	R/S	No	[348,361]
	India	Kochi	ND	5000	ND	No	[351]
	China	RFCq1	ND	>20,000	ND	ND	[272]
	Thailand	Wat Pikul	ND	>125,000	ND	ND	[350]
	Tunisia	TUNIS	ND	~750	R/S	Yes	[143]
	Brazil	Coque	ND	~10	ND	Yes	[379]
		Recife	*cqm1_REC-D16_*/*cqm1_REC-D25_*	3–6	ND	No	[359]
	USA	Chico	ND	687	ND	ND	[353]
		Salt Lake	ND	>20,000	ND	ND	[354]
Laboratory	USA	GEO	*cpm1_GEO_*	>100,000	R/A	No	[157,355]
		L-SEL	ND	37	ND	ND	[380]
	Brazil	R2362	*cqm1_REC_*	>100,000	R/A	No	[149,356]
		RIAB59	*cqm1_REC_*	~40,000	R/A	No	[356,358]
		REC	*cqm1_REC_*	>3425	R	No	[360,363]
		REC-2	*cqm1_REC-2_*	>3475	R	No	[360,363]
	China	RLCq2/IAB59	*cqm1_REC_*	>100,000	R/A	No	[356]
		RLCq1/C3-41	*cqm1R*	>100,000	R/A	No	[272,362]

(a) *Culex pipiens*
*quinquefasciatus* or *Culex pipiens pipiens.* (b) Resistance ratio at LC50 (LC for larvae from a test colony/LC for larvae from a reference colony). (c) Inheritance of resistance: R—Recessive; A—Autosomal; S—Sex-linked. ND: Not Determined.

## 5. Impact on Life Traits

Mosquitoes can be exposed to a variety of stress factors in the environment, including insecticides, and they display mechanisms to overcome toxicity caused by such agents. However, they might be costly and impact life traits [381]. The most critical impact might occur when insects are selected for resistance, and this phenotype can be associated with important biological fitness costs, as was widely reported for resistance to chemical insecticides [382,383]. The fitness reduction may be caused by pleiotropy in the resistance genes themselves or genes closely linked by a “hitchhiking” effect [384]. The action of the Bt toxins in pest insects has been extensively assessed, showing that several biological parameters can be affected [385]. The influence of microbial larvicides on the life traits of mosquitoes has been scarcely studied, and this section aims to present a summary of the available data (Table 5).

Some laboratory-selected colonies resistant to *L. sphaericus* have been investigated. The first *Cx. p. quinquefasciatus* colony studied, which displayed a moderate level of resistance (RR ~ 31- and 37-fold), showed a pronounced reduction in fecundity and fertility [386]. However, analysis of other colonies with higher levels of resistance did not correlate with critical biological fitness costs associated with those phenotypes. An insect colony highly resistant to the 2362 strain (RR > 100,000), showed statistically significant lower fecundity and fertility, but those changes were discrete compared to the susceptible counterparts [387]. Another *Cx. p. quinquefasciatus* laboratory-selected colony highly resistant (RR ≈ 40,000) to the *L. sphaericus* IAB59 strain [360] did not display any significant differences in terms of fertility, fecundity or pupal weight compared to the susceptible individuals [359]. These studies indicated that *L. sphaericus* resistance is not directly associated with the significant biological fitness cost in the development of resistant individuals, at least under laboratory conditions. In the case of agricultural insect pests that developed resistance to Bt toxins, there are reports of discrete impacts on biological fitness costs [388,389,390,391], as well as reports showing high biological fitness costs that could impair the maintenance of the insect colonies [392,393].

In the case of *Cx. pipiens’* resistance to *L. sphaericus* Bin toxin, the resistance is often associated with the lack of expression of the toxin receptor Cqm1 α-glucosidase (see Section 4). In two highly resistant colonies, it was observed that the lack of Cqm1 did not impact the total α-glucosidase activity in the larvae midgut [365]. This study suggested that the expression of another α-glucosidases, paralogs of the Cqm1 protein in larvae, could compensate for the lack of Cqm1. This could explain why the Bin resistance associated with the lack of Cqm1 does not provoke a major biological fitness cost. A similar situation was shown for *Trichoplusia ni* larvae resistant to the Cry1Ac toxin from Bt, which displayed a reduced expression of the APN1, which is one receptor of this toxin whose biological function was compensated by the upregulation of APN6 [394]. These studies support the hypothesis that some resistance alleles are not necessarily linked to crucial adverse effects on biological fitness [359,395]. Indeed, several *Cx. pipiens* colonies resistant to the Bin toxin from *L. sphaericus* were stably maintained for several years under laboratory conditions [143,357,358,359,362,365].

For Bti, such investigation requires a different approach, as there are no reports on resistance to Bti crystals; therefore, the fitness of mosquitoes associated with this specific condition has not been investigated. Despite this, the potential effects of Bti exposure on mosquitoes subjected to a laboratory selection for several generations, or to a short bioassay exposure time (24 h or 48 h), have been assessed. It was observed that mosquitoes continuously exposed to Bti, or to individual toxins from Bti, show some level of resistance to these individual toxins, for instance, an *Ae. aegypti* strain exposed to Bti for 22 generations that did not display resistance to Bti but showed low resistance to some individual Bti toxins (35-fold for Cry4Aa, 11-fold for Cry4Ba) also showed reductions in fertility, in larval viability and an increased larval development time, while adult size, sex ratio, hatching time, longevity, and survival were not changed compared to non-treated individuals [342]. Other studies reported both advantages and disadvantages of some biological traits of *Ae. aegypti* and *Anopheles coluzzii* due to the exposure of these larvae to sublethal doses of Bti [396,397,398]. Although exposure to Bti crystals does not result in the development of insect resistance to the crystals, it is still important to investigate other effects that Bti may induce in the exposed larvae.

In this scope, considering that *L. sphaericus* and Bti are entomopathogenic bacteria whose action depends on the ingestion of crystals/spores by larvae, recent studies have evaluated their impact on the gut microbiota. Indeed, Bti can alter the gut microbiome of *Ae. aegypti* since treated larvae were characterized by a lower bacterial diversity, compared to untreated individuals [399]. The interaction of Bt toxins with the midgut microbiota and the immune system of the insects was recorded, as reviewed by Li et al. [400]. The microbiota can play a major role in the antiviral response of mosquitoes, either by secreting antiviral or proviral molecules or by modulating the immunity response [401,402,403,404]. Some studies have shown alterations in the susceptibility to arbovirus or protozoa in mosquitoes exposed to sublethal doses of *L. sphaericus* or Bti [404,405,406,407]. Therefore, a broader analysis of the potential impact of *L. sphaericus* and Bti on mosquito biology is required to assess the consequences of their use beyond the issue of resistance onset.

**Table 5 toxins-13-00523-t005:** Assessment of biological parameters of mosquitos exposed to *Lysinibacillus sphaericus-* and *Bacillus thuringiensis* svar. *israelensis-*(Bti) based-larvicides.

Larvicide	Specie	Exposure	RR ^(a)^	Parameters ^(b)^		Reference
				Assessed	Altered	
*L. sphaericus* 2362	*Culex pipiens quinquefasciatus*	80 generations	37	FC, FR, DT, SR	FC, FR	[377]
		46 generations	>100,000	FC, FR, DT, ER	FC, FR, DT	[379]
*L. sphaericus* IAB59	*Cx.p. quinquefasciatus*	72 generations	~40,000	FC, FR, PW	None	[357]
*L. sphaericus* 2362	*Anopheles dirus*	48 h	NA	SU-*Plasmodium yoelii*	SU-*P. yoelii*	[399]
Bti	*Culex pipiens pipiens*	20 generations	2.7	FC, LN, TBD	FC	[378]
	*Aedes aegypti*	22 generations	2.0	AS, DT, EV, FC, FR, LN, SR, HT	DT, FR, FC	[340]
		48 h	NA	AS, DT, FC, SV	AS, DT, FC,	[388]
				PS-DENV	None	[388]
		24 h	NA	DT, FC, LN, SR	DT, LN, SR	[389]
	*Anopheles coluzzii*	48 h	NA	AS, FC, LN	AS, LN	[390]
	*Ae. aegypti*	24 h	NA	SU- CHIKV, DENV	SU-DENV	[398]
Bt	*Ae. aegypti*	48 h	NA	SU-DENV, ZIKV	None	[397]

(a) RR: resistance ratio, NA: not applicable. (b) FC—fecundity, FR—fertility, DT—development time, SR—sex ratio, ER—emergence ratio, PW—pupal weight, SU—susceptibility, LN—longevity, TBD—time blood digestion, AS—adult size, HM—haematophagy, EV—egg viability, HT—hatching time, DENV—dengue virus, ZIKV—Zika virus, SV—survival, CHIKV—chikungunya virus, Bt—*Bacillus thuringiensis*.

## 6. Final Remarks

Bti and *L. sphaericus* crystals remain the most powerful and selective insecticidal compounds, available to date, with proven field effectiveness for controlling dipteran species relevant to public health. Recent findings on their mode of action, more specifically on the mechanism of synergistic action of the toxins from both bacteria and the new insights of their interaction with the midgut cells, can be exploited in the future to confer advantages such as broader spectra of action, or to reduce the risk of resistance selection and to improve the persistence under field conditions. Such advancements allied with improved operational practices will allow the evolution of the use of these larvicides from single control agents to their adoption as part of more effective integrated control programs. In addition to the effectiveness of the toxins currently available, these entomopathogenic bacteria also represent opportunities to develop new and/or improved toxins able to display better activities and play an outstanding role in the future of mosquito control.

## Figures and Tables

**Figure 1 toxins-13-00523-f001:**
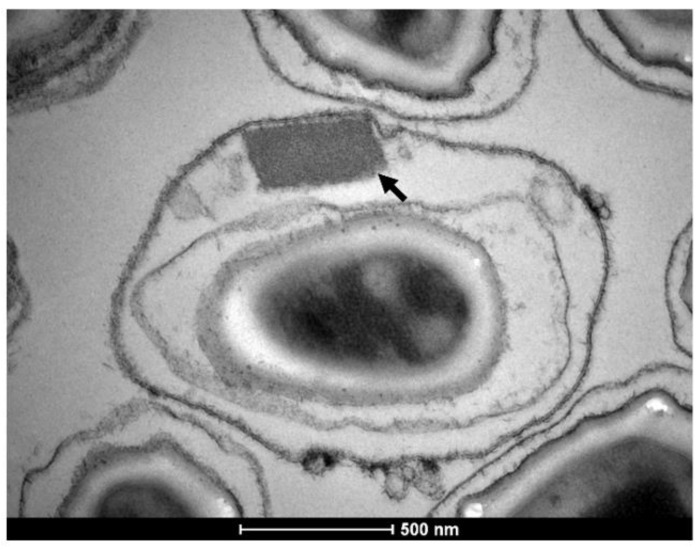
*Bacillus thuringiensis* svar. *israelensis*, 4Q-7 acrystalliferous strain, transformed line expressing the Binary protoxin crystal from the *Lysinibacillus sphaericus* 2362 strain. The arrow points to the crystal. Micrograph kindly provided by Dr. Antônio Pereira-Neves.

**Figure 2 toxins-13-00523-f002:**
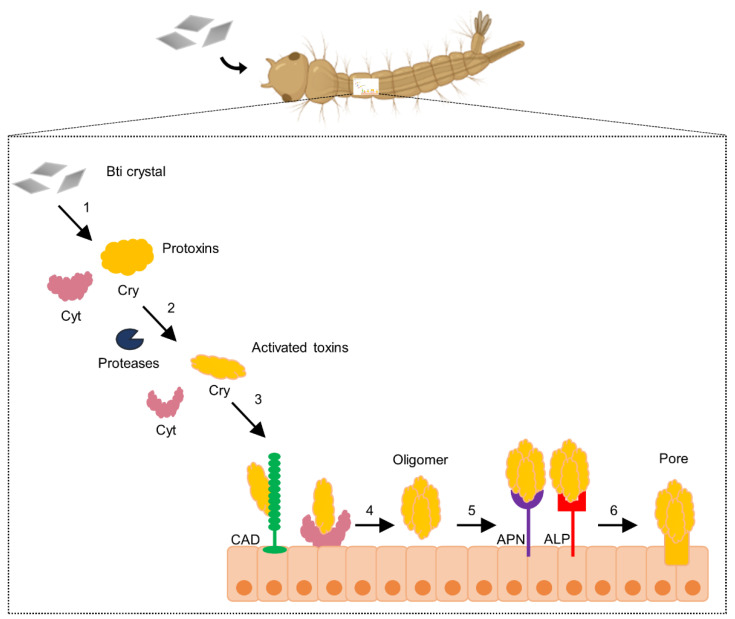
Schematic representation of the mechanism of action of Cry and Cyt toxins from *Bacillus thuringiensis* svar. *israelensis* in mosquito larvae. Crystals ingested by larvae are solubilized in the alkaline pH of the gut lumen (1). The protoxins are activated into toxins by proteases (2); and the Cry toxins can interact with a cadherin or with Cyt1Aa, which also act as a receptor (3); promoting Cry oligomerization (4). This oligomer binds with high affinity to midgut-bound receptors such as aminopeptidases-APN and alkaline phosphatase-ALP (5) and is inserted into the membrane, forming pores (6) that breakdown the cells and kill the larvae. Representation of larvae was created with Biorender.com.

**Figure 3 toxins-13-00523-f003:**
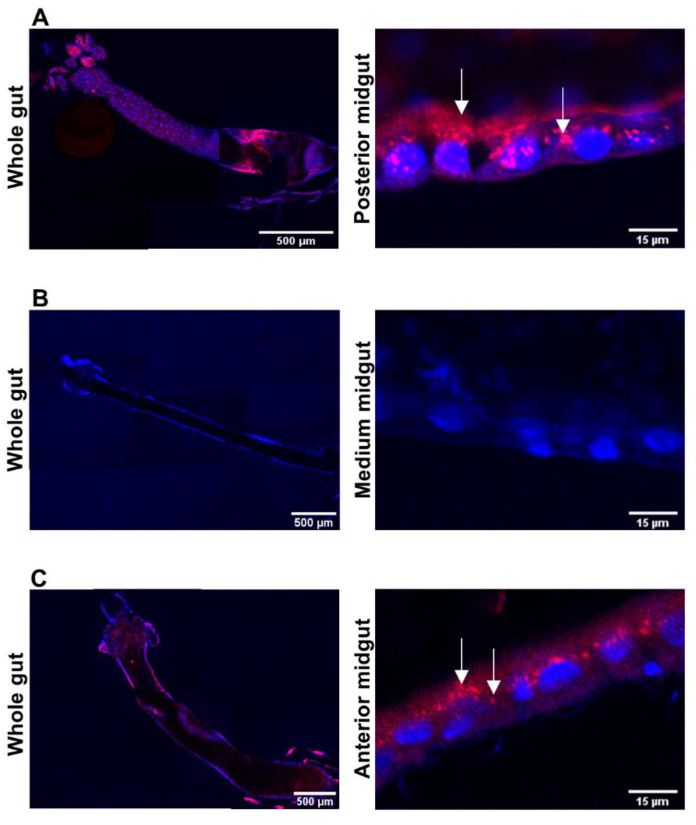
In vivo localization of the labeled Alexa546-Binary (bin) toxin, administrated alone or with the Cyt1Aa toxin (unlabeled) in the midgut of mosquito larvae. (**A**) *Culex quinquefasciatus* treated with Bin; (**B**) *Aedes aegypti* treated with Bin; (**C**) *Ae. aegypti* treated with Bin and Cyt1Aa. Larvae were treated with toxins, processed for microscopy, nucleus were stained with DAPI and labeled Bin toxin (red) was observed with a confocal laser scanning microscope. Arrows point to the Bin toxin binding to cell membrane and internalized into the cell. Figure adapted from [78].

**Figure 4 toxins-13-00523-f004:**
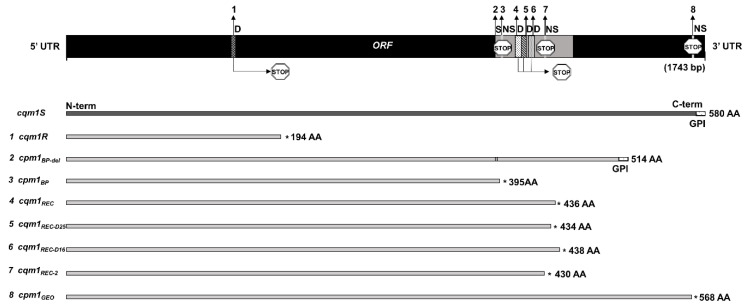
Representation of the open reading frame (ORF) of the *cpm1*/*cqm1* gene (1743 nucleotides-nt) and eight polymorphisms associated to resistance to the Binary toxin that were independently identified. The predicted translated proteins resulting from the *cqm1* susceptible (S) and the polymorphic alleles (1–8) are shown below. 1/ Deletion (D) of a cytosine at position 445 and creation of a premature stop codon downstream. 2/ Aberrant splicing (S) that caused the deletion of 66 residues (V393-Q458). 3/ Nonsense mutation (NS) and creation of a premature stop codon (Gln396Stop). 4/ Deletion of 19-nt. 5/ Deletion of 25-nt encompassing the previous deletion. 6/ Deletion of 16-nt. The deletions from the alleles 4-5-6 create a premature stop codon at the same position. 7/ Nonsense mutation and creation of a premature stop codon (Trp431Stop). 8/ Nonsense mutation and creation of a premature stop codon (Leu569Stop). (*) Truncated proteins without glycosylphosphatidylinositol (GPI) Anchor.

## Data Availability

Not applicable.

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
