# Peer review of "Bacterial Toxins Active against Mosquitoes: Mode of Action and Resistance"

_toxins, 2021, doi:10.3390/toxins13080523_

Round 1

Reviewer 1 Report

Nice review!

Author Response

Reviewer 1

Query. English language and style are fine/minor spell check required

Answer: Our manuscript, before submission, was verified for English editing by https://www.mdpi.com/authors/english  (the certificate is attached) and in this R1 version we corrected other mistakes, all changes are tracked.

Reviewer 2 Report

This review provides general information on bacterial toxins currently used as larvicidal agents in mosquito control programs. This information is abundantly available in the recent literature hence it is difficult to understand what important and novel findings were obtained after the review. E.g., the authors imply that the specificity in the mode of action could counteract the resistance issue. It is widely agreed that resistance development is not the current concern for these bioinsecticides but their environmental and food web-related effects on the ecosystem (C.A. Brühl et al. / Science of the Total Environment (2020)). The authors have completely overlooked this parameter in the current review article. I strongly believe that citing this reference would add value to the current manuscript. 

In abstract: L 4-5: This is a very vague statement especially without any environmental data in the manuscript. To justify this, it would be interesting to expand the ‘resistance’ section with the subheadings and describe the environmental effects of prolonged use of Bti / L. Sphaericus

Also as per the literature,  (Charles, J. F., & Nielsen-LeRoux, C. (2000). Mosquitocidal bacterial toxins: diversity, mode of action and resistance phenomena. Memorias Do Instituto Oswaldo Cruz, 95, 201-206.) It is recommended to use these bioinsecticides as a part of integrated vector control programs rather than their individual large-scale use. What is the author’s conclusion? Should be added to final remarks.  

Adding novelty and specific conclusions/remarks/recommendations would certainly add value and acceptance to the current manuscript. Otherwise, it will be just a piece of similar information available in the literature.

Author Response

ANSWERS TO REVIEWS  

Reviewer 2

Query. English language and style are fine/minor spell check required

Answer: Our manuscript, before submission, was verified for English editing by https://www.mdpi.com/authors/english  (the certificate is attached) and in this R1 version we corrected other mistakes, all changes are tracked.

Query This review provides general information on bacterial toxins currently used as larvicidal agents in mosquito control programs. This information is abundantly available in the recent literature hence it is difficult to understand what important and novel findings were obtained after the review.

Answer: In this manuscript we present the most recent published data on the mode of action and synergism of the major bacterial toxins against mosquitoes. We also present resistance issues and a complete explanation about the use of these toxins worldwide in the recent years. With all these data we have generated a comprehensive and updated review. The main difference with previous reviews is the critical comparison of Bti and L. sphaericus toxins in the same publication presented in this manuscript, including important new data reported in the last five years that has improved our understanding about these insecticidal proteins and their high potential for mosquito control in the future years.

Query E.g., the authors imply that the specificity in the mode of action could counteract the resistance issue. It is widely agreed that resistance development is not the current concern for these bioinsecticides but their environmental and food web-related effects on the ecosystem (C.A. Brühl et al. / Science of the Total Environment (2020). The authors have completely overlooked this parameter in the current review article. I strongly believe that citing this reference would add value to the current manuscript. 

Answer: We do not agree with the reviewer statement that “it is widely agreed that resistance is not the current concern”. Resistance is a problem that we will certainly face in the future and we need to take it seriously. In addition, new available data from the literature show this is indeed an important issue, as well as safety aspects. For this reason, we included the topic “resistance” where we updated the most recent and important information and we presented it in a specific section. Besides, understanding the molecular basis for the lack of resistance to these toxins could give insights for countering resistance for Cry toxins in agricultural pests. On the other hand, we also agree with the reviewer that the topic “environmental and food web-related effects on the ecosystem” was not covered in this review and we have corrected that in the new version of the manuscript. We have included the work of Bruhl et al. 2020, which is an excellent review on this topic, as suggested by the reviewer. We tried to provide to the readers a general view of safety issues and included key references in the new R1 version, as follows:

Line 542 (tracked version without correction marks)

Resistance and safety of larvicidal compounds to mosquito control are prominent issues and they need to be continuously assessed. This section summarizes results from studies that have investigated Bti resistance and also the reports of L. sphaericus resistance, which was already detected. The environmental and human safety issues of the major insecticidal toxins produced by Bti and L. sphaericus have been studied since the early characterization of these entomopathogenic bacteria (e.g. Aly and Mulla 1987, Karch et al, 1990, Lacey & Mulla, 1990, Siegel & Shaduck 1990, Lacey & Merrit 2003) and multiple reports have been published since then. Detailed environmental assessments have been conducted regarding to Bti applications for several decades under the light of actual regulation for the use of biocides in Europe (Pocquet et al 2014, Bruhl et al 2020). In this scope, Bruhl et al. 2020 published a complete review focused to the description of Bti persistence and its environmental impact, including direct effects on the non-target organisms and indirect effects related to the food-chain. The authors presented a detailed analysis and highlight caution regarding to the use of Bti in environmental protected areas, as well as the need of improved monitoring strategies of such effects and adoption of alternative control measures for such habitats. Among some critical issues we can mention the persistence of Bti spores in the soil and its potential impact in microbiota. A recent study analyzed the possible impact of multiple Bti applications in the soil of Riparian wetlands of Switzerland on the population of Bacillus cereus, but no direct correlation was found (Schneider et al. 2017). In terms of safety to other organisms, studies assessing the Bti impact on chironomides, a key element in the food-web chain, showed that the actual criteria of the biocide regulation used in Europe could be underestimated (Kastel et al. 2017, Allgeier et al 2019). It is worth noting that, in some scenarios, chironomides can also be a target species. Some initial assessments of the combined set of Bti and L. sphaericus crystals on non-targets organisms have also been investigated (Duguma et al. 2017, Derua et al. 2018). To date, Bin crystals and Bti crystals are still considered as safe compounds that effectively control several dipteran species larvae of medical importance. However, improved safety assessments should be continuously performed to deep our knowledge about their potential ecological implications, in particular, focusing their use in environmental-sensitive areas.

Query: In abstract: L 4-5: This is a very vague statement especially without any environmental data in the manuscript. To justify this, it would be interesting to expand the ‘resistance’ section with the subheadings and describe the environmental effects of prolonged use of Bti / L. Sphaericus

Answer: As described above, this aspect and key references were included in R1.

Query: Also as per the literature (Charles, J. F., & Nielsen-LeRoux, C. (2000). Mosquitocidal bacterial toxins: diversity, mode of action and resistance phenomena. Memorias Do Instituto Oswaldo Cruz, 95, 201-206.) It is recommended to use these bioinsecticides as a part of integrated vector control programs rather than their individual large-scale use. What is the author’s conclusion? Should be added to final remarks.  

Answer: Yes we agree with the reviewer, please note that in the section 3 of the manuscript the recommendation to use these bioinsecticides as a part of integrated vector control programs rather than their individual large-scale use” is now clearly presented. This section provides several examples of recent field trials in which Bti and L. sphaericus have been used associated and integrated with other tools. This aspect was also reinforced in the final remarks.

Line 472 (tracked version without correction marks)

It is important to highlight that, in the past Bti and L. sphaericus larvicides were used as single control tools in mosquitoes or black-flies control programs that showed effectiveness in a range of scenarios while, nowadays, they are used as part of integrated measures (Charles & NielsenLeRoux, 2000).

Query: Adding novelty and specific conclusions/remarks/recommendations would certainly add value and acceptance to the current manuscript. Otherwise, it will be just a piece of similar information available in the literature.

Answer: The final remarks were modified as follows.

Line 786 (tracked version without correction marks)

Bti and L. sphaericus crystals remain the most powerful and selective insecticidal compounds, available to date, with proven field effectiveness for controlling dipteran species relevant to public health. Recent findings on their mode of action, more specifically on the mechanism of synergistic action of the toxins from both bacteria and the new insights of their interaction with the midgut cells, can be exploited in the future to confer advantages such as broader spectra of action, or to reduce the risk of resistance selection and to improve the persistence under field conditions. Such advancements allied with improved operational practices will allow the evolution of the use of these larvicides from single control agents to their adoption as part of more effective integrated control programs. In addition to the effectiveness of the toxins currently available, these entomopathogenic bacteria also represent opportunities to develop new and/or improved toxins able to display better activities and play an outstanding role in the future of mosquito control.

Reviewer 3 Report

The article "Bacterial toxins active against mosquitoes: mode of action and resistance" proposes a review of the toxins of the bacteria Bti and Lysinibacillus sphaericus from the molecular scale - structure and mode of action - to the description of the resistances induced experimentally or observed in nature to these toxins.

This review provides a longitudinal view of these entomotoxic agents at different scales.

It is coherent, well balanced, and comprehensive.

Author Response

Reviewer 3

Query. English language and style are fine/minor spell check required.

Answer: Our manuscript, before submission, was verified for English editing by https://www.mdpi.com/authors/english  (the certificate is attached) and in this R1 version we corrected other mistakes, all changes are tracked.

Round 2

Reviewer 2 Report

Authors have substantially revised the manuscript.

Minor comment,

Below response from authors should be modified and included in the abstract. This will give readers a clear idea about the status of the review.

"Answer: In this manuscript we present the most recent published data on the mode of action and synergism of the major bacterial toxins against mosquitoes. We also present resistance issues and a complete explanation about the use of these toxins worldwide in the recent years. With all these data we have generated a comprehensive and updated review. The main difference with previous reviews is the critical comparison of Bti and L. sphaericus toxins in the same publication presented in this manuscript, including important new data reported in the last five years that has improved our understanding about these insecticidal proteins and their high potential for mosquito control in the future years."

Author Response

Answer for R2. The abstract was modified following the suggestion

Abstract: Larvicides based on the bacteria Bacillus thuringiensis svar. israelensis (Bti) and Lysinibacillus sphaericus are effective and environmentally safe compounds for the control of dipteran insects of medical importance. They produce crystals that display specific and potent insecticidal activity against larvae. Bti crystals are composed of multiple protoxins: three from the three-domain Cry type family, which bind to different cell receptors in the midgut, and one cytolytic (Cyt1Aa) protoxin that can insert itself into the cell membrane and act as surrogate receptor of the Cry toxins. Together, those toxins display a complex mode of action that shows a low risk of resistance selection. L. sphaericus crystals contain one major Binary toxin that display an outstanding persistence in field conditions, which is superior to Bti. However, the action of the Bin toxin based on its interaction with a single receptor is vulnerable for resistance selection in insects. In this review we present the most recent data on the mode of action and synergism of these toxins, resistance issues and examples of their use worldwide. Data reported in the last years improved our understanding about the mechanism of action of these toxins, showed that their combined use can enhance their activity and counteract resistance, reinforcing their relevance for mosquito control programs in the future years.